# Dynamic-quenching of a single-photon avalanche photodetector using an adaptive resistive switch

Jiyuan Zheng [1,2 ✉], Xingjun Xue[3], Cheng Ji[1], Yuan Yuan[3], Keye Sun[3], Daniel Rosenmann[4], Lai Wang [2,5], Jiamin Wu[2,6], Joe C. Campbell [3] & Supratik Guha[1,7]

One of the most common approaches for quenching single-photon avalanche diodes is to use a passive resistor in series with it. A drawback of this approach has been the limited recovery speed of the single-photon avalanche diodes. High resistance is needed to quench the avalanche, leading to slower recharging of the single-photon avalanche diodes depletion capacitor. We address this issue by replacing a fixed quenching resistor with a bias-dependent adaptive resistive switch. Reversible generation of metallic conduction enables switching between low and high resistance states under unipolar bias. As an example, using a Pt/$Al_2O_3$/Ag resistor with a commercial silicon single-photon avalanche diodes, we demonstrate avalanche pulse widths as small as ~30 ns, 10× smaller than a passively quenched approach, thus significantly improving the single-photon avalanche diodes frequency response. The experimental results are consistent with a model where the adaptive resistor dynamically changes its resistance during discharging and recharging the single-photon avalanche diodes.

[1] Pritzker School of Molecular Engineering, the University of Chicago, Chicago, IL 60637, USA. [2] Beijing National Research Center for Information Science and Technology, Tsinghua University, Beijing 100084, China. [3] Electrical and Computer Engineering Department, University of Virginia, Charlottesville, VA 22904, USA. [4] Center for Nanoscale Materials, Argonne National Laboratory, Argonne, IL 60439, USA. [5] Department of Electronic Engineering, Tsinghua University, Beijing 100084, China. [6] Department of Automation, Tsinghua University, Beijing 100084, China. [7] Material Science Division, Argonne National Laboratory, Argonne, IL 60439, USA. ✉email: zhengjiyuan@mail.tsinghua.edu.cn

Semiconductor p–n junction based single-photon avalanche diodes (SPAD)[1] are attractive as a compact, efficient, and room-temperature technology for applications that include three-dimensional imaging and ranging using time-of-flight methods (LIDAR for autonomous driving[2], gesture recognition[3], 3D scanning[4], quantum communications[5–7], and medical fluorescence monitoring[8]). They operate in the Geiger mode, where the device is reverse biased at a voltage beyond the avalanche breakdown point[1]. The avalanche can be self-sustaining, and a quenching circuit is, therefore, necessary to terminate the multiplication process and reset the device. Passive quenching, the simplest approach, involves adding a sufficient large resistor in series with the SPAD, which, while enabling fast quenching of the avalanche current, results in long reset times because of RC timedelays[9] in the recharging of the SPAD depletion capacitor. Although the RC time constant can be reduced by decreasing the optical sensing area of the detector to reduce junction capacitance (C), it compromises sensitivity[10]. Alternatively, a more complex active or gated mode quenching circuitry can bypass this problem[1]. For example, a variable load quenching circuit (VLQC) can provide the advantages of passive quenching (e.g., suppressing the afterpulsing effect) while greatly reducing the reset time[11,12]. Generally, an MOS transistor controlled by a logic circuit is used to provide the quenching resistance. After the avalanche is sufficiently quenched, the MOS transistor is switched on to accelerate the recharging process. The key point of this idea is to make the quenching resistance large during the quenching and switch it to a low value during the recharging. In this work, we propose and demonstrate a new and scalable approach that accomplishes the same functions as VLQC while retaining the simplicity of passive quenching. The idea is to use a dynamic adaptive resistive switch (ARS), a material whose resistance changes reversibly as a function of the bias across it (thereby obviating the need to alter the resistance using active circuitry), we find one can significantly improve the single-photon detection response times. Under low bias, the ARS possesses a high resistance, but upon applying a bias beyond a certain threshold (on voltage), the resistance drops due to the creation of filamentary conducting paths within the film[13–15]. In some materials, the resistance changes are reversible. There is a return to the initial high resistance state when the bias is reduced below a lower off threshold voltage with the same polarity. When connected in series with the SPAD, this can lead to the adaptive resistor presenting a high resistance during the SPAD discharge process (leading to fast quenching) and a low resistance during recharging (leading to fast recharging). This is accomplished without reducing device area and therefore without compromising SPAD sensitivity. These materials have been widely studied in the context of semiconductor memory technology and as selector switches for cross-bar memories[13,15–23]. A class of these materials consists of dielectric metal oxides such as $HfO_2$ and $Al_2O_3$ (used in this work) placed between an inert electrode (typically Pt) and an electrode of a diffusing metal species (we used Ag), which creates the conducting filament across the $Al_2O_3$ layer by bias-dependent metal diffusion. When the bias is removed, the filament dissolves[15]. We refer to these materials as adaptive resistive switches (ARS) in the context of their use as dynamic quenching elements for SPADs. The resistive state transition in such filamentary resistors has been studied extensively, with switching times reported in the sub-ns level to a few ns level (see Menzel et al.[24] and references [5–19] therein, for instance). Such filamentary devices can achieve a large resistance ratio of $10^7–10^{10}$ under $I–V$ measurement[15,25–28]. In a recent conference abstract, we discussed the method to reduce the recovery time of a SPAD[29] using the resistive switch. However, in this paper, this novel adaptive quenching method is now fully demonstrated.

## Results

**Measurements of avalanche pulse shape via quenching based on ARS**. A schematic of the charging and discharging process of the SPAD in series with a resistor and the experimental system setup used for single-photon avalanche detection are shown in Fig. 1. Details of the experimental measurements are provided in the "Methods" section, along with further information regarding the quenching mechanism. A commercial Si SPAD (Hamamatsu S14643-02) with a sensing diameter of 200 μm was used to detect single photons. The quenching resistors (whether ARS or fixed, passive resistors) were connected in series with the SPAD. A periodic single-photon pulse with a 1 MHz repetition rate incident on the SPAD to trigger an avalanche, and the current flowing through the SPAD derived directly via readout from the oscilloscope. Details of the measurement setup are described in the "Methods" section.

The ARS is a 5 nm $Al_2O_3$ dielectric layer sandwiched by a Ti (5 nm)/Pt (50 nm) bottom electrode and an Ag (10 nm)/Au (50 nm) top electrode. The top and bottom electrodes (each 500 nm wide) are orthogonal, leading to a cross-bar device geometry. Fabrication details are provided in the "Methods" section.

When the ARS is used as a quenching resistor, the typical single-photon triggered avalanche pulse shape (current flowing through the SPAD) is shown in Fig. 2a as the blue curve. Four inflection points are marked in Fig. 2a as (A, B, C, D). The driving voltage of the laser is shown as a red curve. As will be compared later, the pulse shape has a significant difference from those observed in our experiments with conventional passive quenching (i.e., with a fixed resistor). For the current trace of Fig. 2a, one possibility is that the current (A → B) rise is attributable to the discharging process of the SPAD, followed by B → C and C → D corresponding to recharging processes. If this were the case, then the quenching resistance during B → C is larger than that during C → D period (since the B → C segment slope is lower than the C → D segment). This would imply that the switch to the low-resistance state of the ARS occurred around point C, and there should then be a significant rise in the SPAD current at C, which we do not observe.

Therefore, discharging is completed before point A and point A is involved in the recharging process. The rise of current at point A is caused by the switching of the resistance for the ARS (as shown in Fig. 2a). This is indeed expected if we consider estimates of the timescales involved: the RC time constant for discharge is ~700 ps for a junction capacitance of 0.7 pF (datasheet) and a diode resistance of 1 kΩ (The estimation of diode resistance is discussed in "Methods" section). It follows that ~90% of the stored energy will be discharged in ~1.6 ns (from Eqs. (1) and (2)). In contrast, resistive switches are known to switch on timescales of ~100 ps to a few ns (c.f. Menzel et al.[24] and the multiple references [5–19] listed therein). Based on these estimates' expectations, we propose that at point A the SPAD has already completed its discharge, and segment A → B is caused by the ARS switching from the off (high-resistance) to the on (low-resistance) state. Segment B → C represents the fast recharging period with the ARS in the on state. When the voltage across the ARS drops below a critical value (the off voltage) at point C, the ARS reverts to the off state, leading to the C → D segment. It should be noted that the SPAD recovery will continue after point D and the increased resistance of the ARS prolongs the process. Thus, in this paper, we refer to the A → D process as the critical recovery process, which is accelerated by the ARS and the duration after D is designated as the recovery tail. In this paper, the crucial point is that the critical recovery process is fast, during which time the voltage restored on the SPAD is sufficient to let

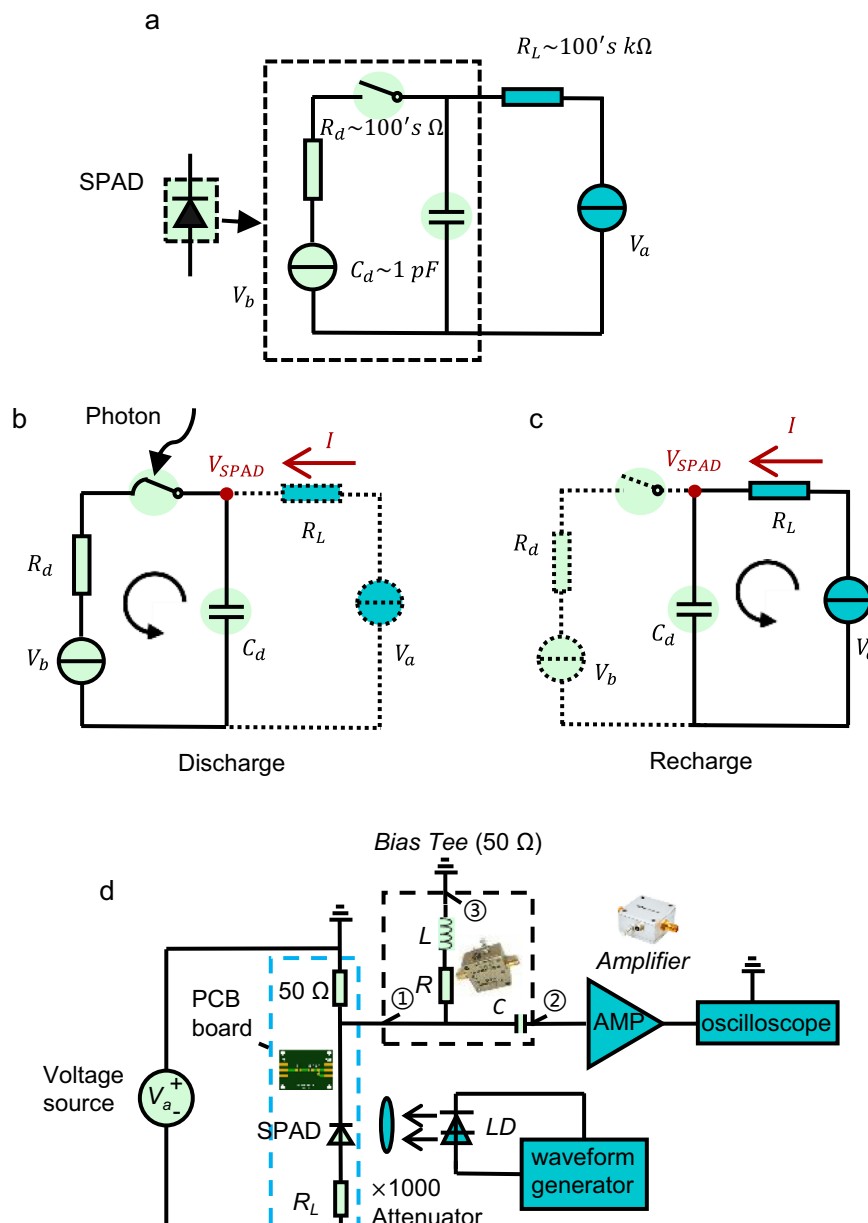

**Fig. 1 Circuit model for passive quenching of the single-photon avalanche photodiode (SPAD) and the experimental setup of the single-photon measurement. a** Equivalent circuit model for passive quenching of the SPAD in series with a quenching resistor. The dashed box is the SPAD, represented by a capacitor, a resistor, a photon switch, and voltage source. **b** The SPAD discharge process when an avalanche is triggered by an incident photon. **c** Recharging process after quenching. **d** Single-photon detection system. Passive quenching is accomplished by connecting a SPAD and a resistor in series. A voltage source (Keithley 2400) is used to reverse bias the SPAD. The single-photon signal is generated by attenuating the optical signal from a laser diode driven by a Keysight 33600A waveform generator. The avalanche response current is determined from the voltage on a 50 Ω readout resistor and the voltage is introduced to an amplifier (Mini-circuits ZFL-1000LN+) and then into an oscilloscope (Rigol DS7024) through a bias tee (Mini-circuits ZFBT-4R2GW+).

the SPAD detect other photons. The critical recovery process dominates the counting speed of the SPAD. Although during the recovery tail process, the SPAD bias is slowly restored, it does not significantly influence the detection efficiency. Since it is hard to sense the exact value of the off-state resistance of the ARS in the serial system during fast quenching and recharging process, it is difficult to precisely determine the recovery and is beyond the scope of this paper.

Further analysis of this behavior via simulations is described later in this paper. A statistical analysis of the critical recovery times taken over 1000 avalanche pulses, and the critical recovery time distribution is shown in the histogram of Fig. 2b. Most pulses have a short critical recovery time (<50 ns), and the averaged critical recovery time is estimated as 30 ns. Since the laser pulse width is 15 ns, and we target single-photon pulses via the use of a 1000× attenuator applied to a ~1000 photon number (average) laser pulse, we cannot rule out the stochastic impingement of multiple (few) photons rather than a single photon only. The detection of one or a few photons is not relevant to the purpose of the current paper, which is to demonstrate the dynamic quenching operation of the ARS. However, this variation may play a role in our observation of the

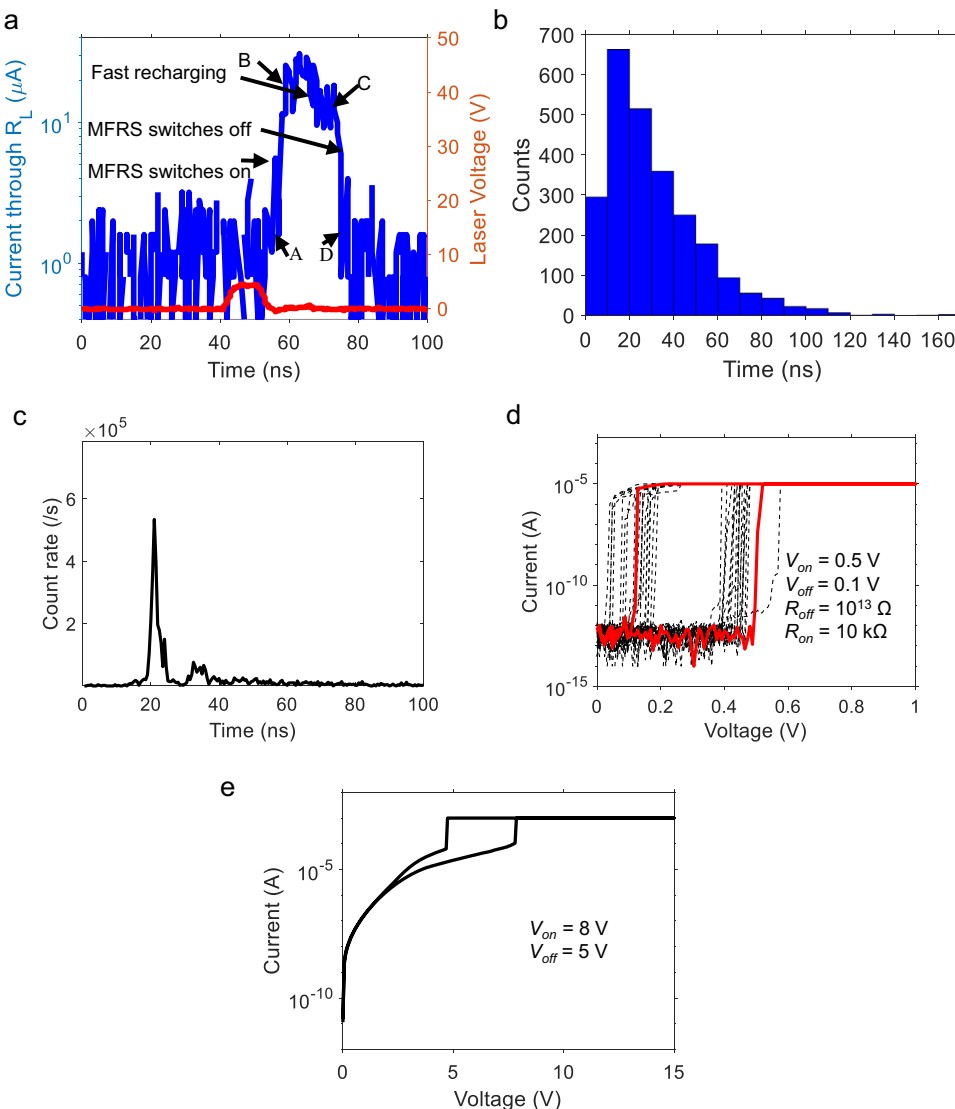

**Fig. 2 Quenching experiment conducted on the adaptive resistive switch (ARS). a** Typical pulse shape of the ARS quenched single-photon avalanche photodiode (SPAD) (blue curve); the red curve indicates the driving voltage of the laser. **b** The statistical distribution of the critical recovery time for the SPAD quenched by the ARS. The average critical recovery time is 30 ns. **c** Jitter performance of the ARS quenched SPAD when operated at 6 MHz repetition rate. **d, e** Current–voltage characteristics of the ARS showing the hysteresis behavior of the resistor, measured: **d** before and **e** after an estimated $10^{10}$ cycles of switching (quenching).

spread in the critical recovery times. In addition, the variation of pulse width in the output of the SPAD quenched by the ARS is also probably caused by the stochastic electro-chemical reaction processes of the ARS filament growth and dissolution.

The jitter performance of the ARS quenched SPAD is calculated from the avalanche output measured by the oscilloscope (Fig. 2c). Details are provided in Supplementary Materials. The threshold for counting is set to be 5 mV, and the sampling time step is set to be 0.5 ns. A sharp and high peak appears at $t \approx 21$ ns (full width at half maxima, FWHM ~1.5 ns), while there is a second small peak located at $t \approx 35$ ns, which is caused by the ARS degradation and will be discussed later. Degradation measurements indicate that the jitter degrades with repeated operation—these results are discussed later in the paper. In our measurements, the modulation bandwidth of the TO-packaged laser (Thorlabs L520P120) compresses the 15 ns pulse width of the drive waveform (voltage monitored as shown in Fig. 2a red curve). The actual current waveform is narrower than the electric pulse, as a result of which the jitter time is much shorter than the

electric input pulse width of the laser (15 ns). The measured jitter indicates that most avalanche responses happen with good timing consistency due to the fast and critical switching of the ARS.

**Hysteresis behavior of ARS**. The quasi-static current–voltage ($I$–$V$) measurement of the ARS following the standard forming treatment at 5 V[30] (and before the quenching experiments) is shown in Fig. 2d. A compliance current (1 mA) is used to restrict the conducting filament thickness to keep the device under a volatile mode (i.e., a reversible return to the high resistance state at $V = 0$)[26,31,32]. As can be seen, the on voltage is ~0.5 V, the off voltage is ~0.1 V, and the off-state leakage is <1 pA. The off-state leakage current flowing through the ARS is lower than the Keysight B1500a semiconductor analyzer discrimination level and is buried by its noise floor as shown in Fig. 2d. Following repeated operations during our experiments, the on and off switching voltages drift upwards, with an increase in leakage current. This can be seen in the $I$–$V$ characteristics of Fig. 2e (measured with

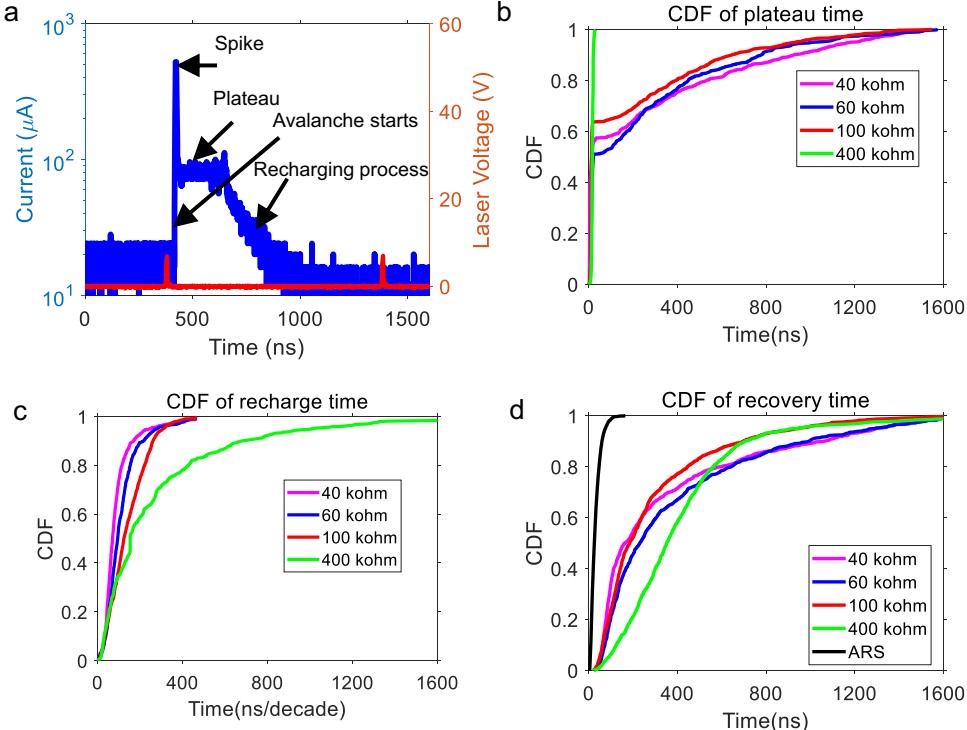

**Fig. 3 Results of quenching experiments conducted using fixed quenching resistors. a** Typical response pulse shape of the 60 kΩ quenched single-photon avalanche photodiode (SPAD) (blue curve) indicating features corresponding to the spike, plateau, and recharging processes (pulse shape for the 40, 100, and 400 kΩ quenched SPAD has similar features); the red curve indicates the driving voltage of the laser. **b, c** Cumulative distribution functions (CDFs) of the plateau and recharge times, respectively, for the 40, 60, 100, 400 kΩ quenched SPAD. **d** CDFs of the total recovery times for the 40, 60, 100, 400 kΩ, and the critical recovery time of the adaptive resistive switch (ARS) quenched SPAD (black).

the same time scale as Fig. 2d), taken after ~10^10 avalanche triggers at periodical operation single-photon signal. The on and off switching voltages have drifted upwards to 8 and 5 V, respectively, and off-state and on-state resistances at this condition are ~400 and ~40 kΩ, as is fitted by the simulation, which will be discussed later. The consequences of this drift in relation to degradation are discussed later. The degradation will need to be improved through materials development and is not unusual for new device development.

**Comparison with conventional passive quenching.** To achieve a comparable (to the ARS-based results) recovery time of ~30 ns using conventional passive resistance quenching, we estimate (using Eq. (3) in the "Methods" section) that the resistance should be ~18.6 kΩ. Here, we have used a junction capacitance of 0.7 pF for the SPAD (datasheet of Hamamatsu S14643-02). However, such a small resistance value is not expected to be adequate to quench the avalanche process[1]. When we used a 30 kΩ resistor in series with the SPAD, the avalanche sustained itself, the high current degraded the SPAD performance, and the avalanche response current became noisy within seconds, followed by burnout of the SPAD. We then carried out quenching experiments on the same SPAD type with quenching resistors of 40, 60, 100, and 400 kΩ: the results are shown in Fig. 3a–d. Different excess bias voltages are used to ensure that the counting rate at low repetition rate is the same ("Methods" section). The measured avalanche pulse response (Fig. 3a) indicates three features: a spike, a plateau, and a recharging process. According to Marano's work[10], the spike originates from the fast charging of the chip quenching resistor's stray capacitance. However, no spike is observed for ARS quenching (Fig. 2a). The spike's absence is

because the ARS is mounted on a TO-5 can package, and the lead pitch (5.08 mm) is much larger than the bottom termination distance of the chip resistor (0.3 mm) used in the conventional quenching method. Since the stray capacitance is parallel to the ARS, the capacitance is inversely proportional to the lead pitch. Thus, the stray capacitance is much smaller and can be ignored. The plateau is due to a sustained avalanche that occurs when the quenching resistance is not large enough[33]. The variation in the plateau's duration is a consequence of the probabilistic nature of the quenching process[1,34]. In Fig. 3b–d, we have compared the plateau time, recharging time, and recovery time for the SPAD passively quenched with 40, 60, 100, and 400 kΩ fixed resistances. As can be seen, in all cases the recovery time is typically ~300 ns or higher (Fig. 3c) for a CDF of >0.75. When the fixed resistance is 30 kΩ or lower, we have observed unreliable quenching of the avalanche so that the operation becomes unreliable. In the case of the ARS, on the other hand, we are able to observe a 10× reduction in response time to ~30 ns. Our passive resistance data suggests that this is not possible if the ARS simply had a fixed resistance, since we have bracketed our fixed-resistance data going down to values at which point the avalanche cannot be quenched. Therefore, we believe that the dynamic switching of the ARS is leading to this lowered quenching resistance for the ARS case. It should be noticed that the typical recovery time for conventional passive quenching of SPADs with large sensing area (diameter > 100 μm) is 500 ns to 1 μs[1]. The use of ARS can significantly improve the response speed of SPADs. To achieve comparable speeds using conventional passive quenching would require reducing the sensing area to around 10 μm, which complicates optical coupling and can result in reduced sensitivity[10,35]. The randomness of the plateau duration originates from the randomness of quenching time. When the quenching resistance is

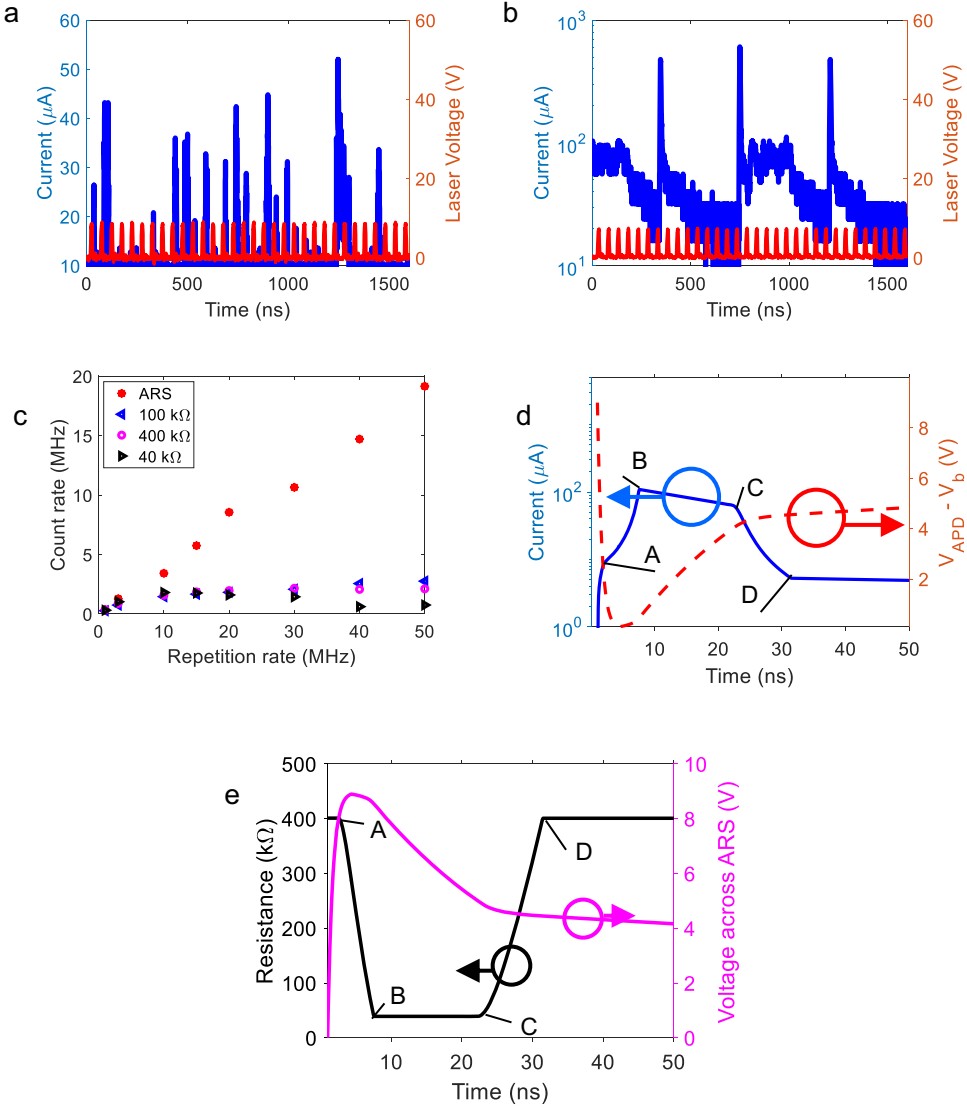

**Fig. 4 Performance comparison between adaptive resistive switch (ARS) quenching and passive resistive quenching of the counting speed of a single-photon avalanche photodiode (SPAD). a, b** Avalanche response to a single photon signal with a 20 MHz repetition rate for a single photon avalanche photodiode (SPAD) quenched by **a** the adaptive resistive switch (ARS) and **b** the 100 kΩ. **c** Counting rate under different repetition rate for SPAD quenched by ARS and fixed resistor. **d, e** Pspice simulations of a SPAD quenched by the ARS showing the variation of the following parameters as a function of time: **d** the avalanche current trace (blue curve) and the voltage across the SPAD (red dashed curve); and **e** voltage across the ARS (pink curve) and the resistance (dark curve).

large enough to quench the SPAD, the time variation is suppressed, and the quenching time is merely dependent on the R-C time constant of the SPAD internal discharging loop. Whereas when the quenching resistance is not large enough, there is a high probability that the quenching resistor does not immediately quench the SPAD, and the avalanche could last until it is self quenched, during which there is significant randomness[33].

The impact of faster critical recovery times is illustrated in high repetition rate (20 MHz) single-photon measurements (Fig. 4a and b). Details of the measurements are presented in the "Methods" section. Representative avalanche responses (across 1.6 µs time windows) are shown both for the adaptive quenching (ARS) and conventional passive quenching (100 kΩ) cases. The red curve indicates the single-photon drive voltage, and the blue curve is the SPAD signal. Statistical analysis of the data was carried out using single-photon response data over 0.4 ms with a time step resolution of 0.4 ns. There are 8000 single-photon pulses involved in the analysis. Details of the analysis are

provided in the Supplementary Materials section. The single-photon counting rate under 20 MHz single-photon repetition rate is 1.8 MHz for conventional passive quenching (100 kΩ) and 8.5 MHz for ARS quenching.

Similarly, the counting rates under different repetition rates ranging from 1 to 50 MHz are calculated and plotted in Fig. 4c for SPADs quenched by ARS and conventional passive quenching. For comparison, 400 kΩ ($R_{off}$ of ARS), 40 kΩ ($R_{on}$ of ARS), and 100 kΩ are used to perform the conventional passive quenching. As is shown in Fig. 4c, the counting rate of the ARS quenched SPAD is significantly higher than the passive quenched SPAD, especially when the repetition rate is large. The results are consistent with the faster critical recovery times of the SPAD measurements with ARS quenching.

However, a slow and continuous increase of the voltage across the SPAD after point D (Fig. 2a) would lead to higher dark counts. In the experiment, the dark count rate of the SPAD quenched by a 100 kΩ resistor is 207 kHz, while that for the

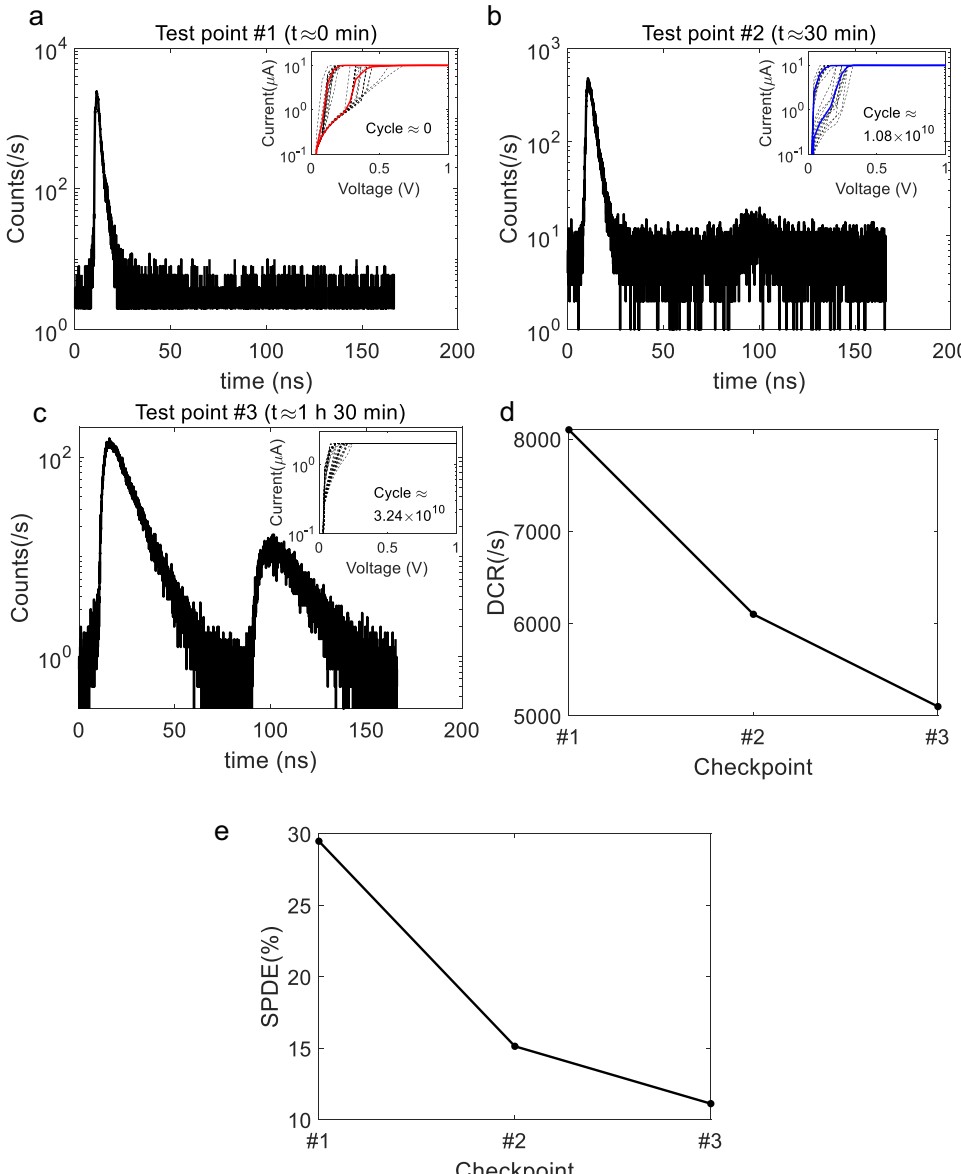

**Fig. 5 The counting performance of single-photon avalanche photodiode (SPAD) quenched by the adaptive resistive switch (ARS) degrades with time.**
**a–c** Counting histograms tested on SPAD quenched by the ARS when the pulse repetition rate is 6 MHz. **a** Checkpoint #1 when $t \approx 0$ min, ARS switching cycle $\approx 0$. **b** Checkpoint #2 when $t \approx 30$ min, ARS switching cycle $\approx 1.08 \times 10^{10}$. **c** Checkpoint #3 when $t \approx 1$ h 30 min, ARS switching cycle $\approx 3.24 \times 10^{10}$. As time increases the SPAD dark count rate **d** and single-photon detection efficiency **e** both decrease.

SPAD quenched by the ARS is 330 kHz. In future work, the stability of the ARS can be improved, so that the off voltage of the ARS is close to 0 V and the recharging process stops right after point D (Fig. 2a).

**Analysis of the ARS quenching of the SPAD.** The narrow avalanche pulse width and fast quenching performance are consistent with a critical change in the resistance of the ARS when quenching the SPAD. The switching mechanism of resistive switches has been extensively studied, as noted earlier. Switching times have been reported in the range of ~100 ps to a few ns[24]. There have been limited reports on the switching behavior of similar resistive switches in series with a diode (or capacitor) and the impact of the capacitor's charging and recharging process. The results reported here are consistent with a switching time on the level of a few ns.

In the following, we discuss the modeling of the ARS quenched SPAD's response via PSPICE simulations using OrCAD Pspice Designer. Details of the analytical model are presented in the "Methods" section. For this simulation, the switching voltages and resistances of the ARS are extracted from the $I$–$V$ measurement results, as is shown in Fig. 5a. The ARS on and off time constants were empirically set at the level of 1 ns.

The simulation results are shown in Fig. 4d and e. In Fig. 4d, the response current is shown in solid blue curve while the excess bias, which is defined as the voltage across SPAD minus breakdown voltage is shown in dashed red curve. The shape of the blue current curve is similar to that observed in the experiment. The SPAD abrupt voltage drop illustrates how the discharge proceeds (1–4 ns). After the discharge, the ARS starts to switch and generates an avalanche pulse output with A → B → C → D periods similar to the experimental results shown in Fig. 2a. Figure 4e shows the voltage across the ARS (purple curve)

and the ARS resistance (dark curve). Using relevant physical parameters (see "Methods" section), the simulations show that with the triggering of an avalanche, the junction capacitor of the SPAD discharges, and the ARS switches from its high (400 kΩ) to low (40 kΩ) resistive states in 4.7 ns (A → B).

It should be noted that point A ($t = 2.47$ ns @Fig. 2a) occurs during the discharging period (1–4 ns), after the voltage across the ARS exceeds 8 V. As a result of switching-on, the current increases, and the recharging process is then accelerated (B → C in Fig. 4d, e). During the fast recharging, the excess voltage across the SPAD increases to 4 V (red dashed curve in Fig. 4d), and the voltage on the ARS is reduced to below 5 V (purple curve in Fig. 4e). As a result, the ARS switches off (C → D in Fig. 4e). The recharging process then decelerates (C → D in Fig. 4d). The shape of the simulated SPAD response is consistent with experimental observations, which means that the ARS switches resistance from the high to the low state during the SPAD discharge and recharge process of the SPAD, thereby significantly reducing SPAD reset times. However, two issues remain unresolved in the model used. First, the magnitude of the current in the simulation peaks at ~100 μA, which is higher than what is observed (10–40 μA). The reason for this is not clear at this moment. Second, our model does not incorporate statistical fluctuations of the avalanche pulse width (Fig. 2b).

## Discussion

We demonstrate in this paper that the ARS accomplishes resistive switching during the avalanche process with the result that the avalanche reset is greatly accelerated compared to a passive resistor. Although the critical recovery time (30 ns) is much longer than the state of the art VLQC method (2–3 ns @sensing area size of 20 μm[11,12]), the ARS quenching method holds significant advantages in suppressing the afterpulsing effect with a large initial quenching resistance (400 kΩ in this work, a few tens of kΩ in ref. [11] (VLQC), and 800 Ω in ref. [12]) and therefore the sensing area size can be much larger (200 μm in this work) and there is no need to design a hold off time, which can be quite long (>20 ns) in the VLQC method. Moreover, VLQC requires more processing complexity. We are offering a neat and simple way to get 10× improved critical recharging speeds in large SPADs with just a swap of the resistor. Based on the working principles analyzed in this paper, this critical recovery time may be further shortened by reducing the on-state resistance (to accelerate the recharging process) and improving the redox speed of the Ag electrode (increasing the critical switching speed of the ARS).

We note that printed circuit board (PCB) interconnects limit our time resolution for the single-photon counting to a few nanoseconds. Hence our measurements cannot probe the sub-ns dynamics of the filamentary devices that have been reported by other workers in similar materials (see Menzel et al.[24] and references [5–19] therein, for instance). However, our approach is adequate for clearly demonstrating the clear benefits of the ARS devices in reducing passive quenching response times to tens of nanoseconds and by a factor of 10× as compared to the passive quenching case.

We now turn to a discussion of the drift in the ARS characteristics, as was noted earlier. A photon counter (HydarHarp 400) was used to obtain a counting histogram of the SPAD quenched by the ARS (details are provided in the "Methods" section). Compared to the measurements of Fig. 2a–c and Fig. 4a–c, we used the same model of SPAD (Hamamatsu S14643-02) but employed a lower overvoltage (2 V) compared to 9 V for the earlier measurements. The light pulse was attenuated to 0.1 photons/pulse. As is shown in Fig. 5, we measured the histogram at three different time points (Checkpoints #1, #2, and

#3). Checkpoint #1 is chosen at the beginning of the measurement when $t \approx 0$ min and the ARS switching cycle ≈0. There is a significant peak, indicating a good timing response. The dark count rate (DCR) is around 8 kHz, and lower than the value in the measurements for Figs. 2 and 4 (330 kHz) due to the smaller overvoltage. The single-photon detection efficiency (SPDE) is 30%. Unipolar I–V hysteresis performance is shown in the inner-plot, representing a stable hysteresis performance. After continuous operation for 30 min, the SPAD performance was measured again (Fig. 5b, Checkpoint #2). Note the appearance of a second peak in the histogram. The DCR decreases to around 6 kHz, and the SPDE decreases to 15%. The I–V hysteresis shown in the inner-plot indicates that the ARS has become leakier. We infer that it is becoming harder for the ARS to be switched off, so it is highly probable that the avalanche could not be quenched, and so the counting rate is suppressed. The second peak is probably caused by the longer restoring time of the ARS. After 1 h 30 min (Checkpoint #3), the second peak has increased in magnitude relative to the first. As The DCR and SPDE decrease to around 5 kHz and 11%, respectively. As seen in the I–V hysteresis curve, the ARS tends to switch to non-volatile mode, indicating that it is much harder for the ARS to be switched off by the unipolar driving voltage.

The FWHM of jitter distributions (first peak) shown in Fig. 5a–c are quantified as 2, 3.6, and 10.3 ns at the three checkpoints over time. The degradation of single-photon detection performance is mainly caused by the limited device endurance. The fast avalanche pulse response is enabled by the switching of the ARS between on and off states. With accumulated photon counting, the ARS becomes leakier and harder to switch off, causing the SPAD performance to degrade. It is well known that lower quenching resistance degrades after-pulsing and jitter performance[1,36]. We anticipate, therefore, that the increased leakage in the ARS will also lead to poor jitter and after-pulsing characteristics. These measurements were not carried out.

It should be noted that the data in Figs. 2–4 for ARS quenched SPAD was collected within a short time (when the I–V performance of Fig. 2d was tested), during which the behavior of the ARS did not change appreciably. Such drift can arise from microstructural changes during the conducting filament formation and dissolution, leading to eventual device degradation.

Detailed studies of the degradation process and statistical evaluation of the fatigue characteristics are the future research subjects and are outside this paper's scope. We note that similar considerations relating to material stability under repeated operation have also been the subject of significant work in developing this class of materials for non-volatile memory applications with high cycling endurance. For instance, studies have shown that endurance can be improved via using alloy electrodes like Ag-Te[37], Ag-Cu[38], inserting Ag diffusion barrier layer[39], area scaling of the device switching region[40], using host materials with stronger chemical bonding among its components[41], nitridation[42]. We anticipate that resistance to such microstructural degradation for the case of the ARS may similarly be achieved by designing optimized electrode, switching structures, adjusting resistor area, new host matrix and electrode materials, and the introduction of solute additives that can retard diffusive processes that exacerbate microstructural fatigue.

Silicon SPADs are technologically relevant for use as fluorescence monitors in biomedical applications. The widest range of such applications are at room temperature due to issues of cost, practicality, and application space. Silicon SPADs can fit this bill since, unlike longer wavelength detectors such as InGaAs and HgCdTe-based SPADs, the Si dark current density is three orders of magnitude lower (compare for instance the Hamamatsu S14643-02 Si and G14858-0020AA InGaAs detectors) at room temperature. Furthermore, a major intent for resistively quenched SPADs is

reduced cost and complexity. This is mostly also applicable to room temperature measurements. Lower temperature applications have a higher cost ceiling at which point fast active quenching circuitry can be incorporated, and there is no need for resistive quenching. Our studies have therefore focused only on room temperature measurements of ARS-based quenching. At lower temperatures, we would expect the ARS switching speed to drop (ref. [43]) and the switching voltage to increase (Huang's work[44]). So, a careful calibration is needed when using ARS to quench the SPAD at low temperatures to enhance the SPAD performance[45].

The afterpulsing probability was estimated by analyzing the oscilloscope data (for 1 and 3 MHz repetition rates) in the following manner. A mimic dual pulse window varying from 0.1 to 100 ns was used to gather the afterpulsing peaks that occurred following the photon-generated pulse. We used 5 and 35 mV as the counting thresholds for the ARS and the fixed resistor quenched SPAD, respectively. The afterpulsing probabilities for the SPAD quenched by the ARS, and the 400, 100, and 40 kΩ, respectively, are 7.6%, 2.45%, 8.6%, and 18.3%. The better performance of the ARS compared to the 40 kΩ fixed resistor is due to the critical switching of the ARS and an averaged large off-state resistance, which quenches avalanche fast and reduces the number of carriers that flow through the avalanche region, leading to a lower probability of afterpulsing[36]. The poorer performance of the ARS compared to the 400 kΩ fixed resistor is likely due to an increase in the probability of switching failure caused by degradation.

It should be noted that although the external drive voltage to the circuit is over 100 V, most of it drops across the SPAD, not the ARS. The maximum voltage drop that develops across the ARS occurs when there is a dynamic change of the voltage across the SPAD junction capacitance, and is limited to a few volts. To protect the ARS from burning, two points should be guaranteed: (1) The overvoltage should not be too large and (2) the external voltage should be loaded and unloaded gradually to prevent an un-expected sudden voltage drop on the ARS (the impendence of SAPD is small when the frequency is large).

The linear model (Eqs. 1–4) used in this work is a simplification for the ARS dynamic behavior. While our simulations fit the avalanche pulse well, we note that there is a discrepancy when fitting the I–V curve due to the assumption of linearity. This is described in the Supplementary section and in Fig. s4. According to Russo's work on the resistance of metallic filamentary switches that are progressively driven by an electric field, the switching speed can be regarded as constant on a short time scale (tens of nanoseconds)[46]. Since the I–V sweep is a long process (>50 ms), we believe the linear model is inadequate for accurately fitting the I–V curve.

The current largest application of passive quenching is SiPM (silicon photomultiplier)[10,47–49]. This work is potentially beneficial for improving the performance of the SiPM by accelerating the recovery speed of avalanche quenching. In addition, the ARS is easy to fabricate and compatible with the Si material system. Therefore, the potential for integration with SPAD arrays is high.

In summary, an avalanche photodetector quenched with a self-adaptive resistive switch (ARS) has been proposed and demonstrated experimentally. We find that this approach led to an avalanche pulse width is at least eight times narrower than the conventional passive quenching method while retaining its approach's simplicity. The experimental data and simulations support our contention that such fast switching is accomplished due to the voltage-dependent resistance of the ARS switch. In response to the bias changes across the ARS during the discharging and charging processes, it presents a high resistance during the SPAD discharge process, drops to a low resistance during the recharge process, and resets to a higher resistance value following the recharge.

## Methods

### Working principles of the SPAD quenching and how the adaptive resistive switch works in the system.

The principle of operation is described using the equivalent circuit of a conventional passively quenched SPAD (Fig. 1a): represented by a photon activated switch, an diode resistance ($R_d$) and a voltage source ($V_b$, representing the avalanche breakdown voltage of the SPAD) and a junction capacitance ($C_d$) as shown in the figure[48]. The SPAD connects to an external voltage source, $V_a$ ($V_a–V_b$ = overvoltage), and an $R_L$ quenching resistor in series. Initially, the SPAD is charged by the external voltage, resulting in voltage $V_{SPAD} = V_a$ applied across $C_d$. Upon absorption of a photon (Fig. 1b), the switch closes, and $C_d$ discharges through the internal loop (i.e., the avalanche is triggered), accompanied by a drop in $V_{SPAD}$. When $V_{SPAD}$ decreases to a value around $V_b$, the avalanche process ends, and the switch reopens (Fig. 1c). The external voltage now charges the SPAD, and the cycle is complete with the SPAD ready for detection when the capacitor is fully charged. The currents $I$ and voltages $V_{SPAD}$ during discharging are derived and given as Eqs. (1 and 2), and during recharging are derived and given as Eqs. (3 and 4):

$$I = (V_a - V_b)(1 - e^{-t/R_d C_d})/R_L \tag{1}$$

$$V_{SPAD} = (V_a - V_b)e^{-t/R_d C_d} + V_b \tag{2}$$

$$I = (V_a - V_b)e^{-t/R_L C_d}/R_L \tag{3}$$

$$V_{SPAD} = -(V_a - V_b)e^{-t/R_L C_d} + V_a \tag{4}$$

The diode resistance ($R_d$) is the serial sum of the resistance in barrier region (the neutral region that current goes through) and space-charge layer. A smaller sensing area and a thicker depletion region would lead to a larger diode resistance. The typical diode resistance is in the range of 100 Ω to a few kΩ[1]. The diode resistance of the SPAD used in this paper is taken to be 1 kΩ since it has a large sensing area (diameter is 200 μm) with a relatively thick barrier region (the quantum efficiency can reach as high as 85% @ 650 nm wavelength).

In SPADs, a large $R_L$ facilitates sufficient quenching and a lowered jitter time in the discharging process[1]. As a result, $R_L$ is typically held at ~100 kΩ[1]. However, as shown in Eq. (4), this high quenching resistance also increases the recharging time significantly due to a high $R_L C_d$ value (since $R_d$ is typical ~100 Ω to a few kΩ[1], the discharging time—Eqs. (1) and (2)—can be ignored compared to the recharging time). Since the probability of an avalanche triggered by newly absorbed photons is very low while the SPAD is being recharged, this longer recovery time limits the SPAD's frequency response when passive quenching is used. What is needed to improve the SPAD's frequency response is a dynamic resistor with a high resistance during the discharge process and a low resistance during recharging.

The ARS device is connected in series with the SPAD (replacing the passive resistor in Fig. 1). For the process to be successful, a dynamic interaction between metallic filament formation kinetics and avalanche quenching needs to occur. When absorbed photons trigger an avalanche in the SPAD, the ARS is in the off-state (high resistance). The SPAD depletion capacitance discharges and the avalanche is quenched when $V_{SPAD} < V_{breakdown}$. Until this point, the ARS resistance should remain in the high resistance state to ensure rapid quenching of the SPAD. Following avalanche termination, the ARS should switch to the low resistance state driven by the voltage built up across it due to the drop in $V_{SPAD}$. This time scale is dictated by the formation of the conductive filament across the oxide due to metal drift under the electric field. The transition to the low resistance state in the ARS, in turn, enables rapid recharging of the SPAD. As the recharging progresses, the voltage across the ARS now decreases, and when it attains a value smaller than the off voltage of the ARS, the conductive filament dissolves. The ARS returns to its high resistance off-state, and the SPAD circuit is reset. The dynamic lowering of the ARS resistance enables rapid resetting of the SPAD circuit.

### ARS fabrication and measurement.

The ARS devices were fabricated on Si wafers covered with 300 nm thermal SiO₂ (see Fig. s1 in the Supplementary section for the experimentally fabricated devices). In this paper, a typical cross-bar architecture[50] is used to form the device geometry. The 500-nm-wide bottom electrode strips were fabricated by electron-beam lithography followed by electron-beam evaporation of a Ti (5 nm)/Pt (50 nm) bilayer thin film and lift-off. Next, an AlOₓ layer was deposited by atomic layer deposition (Veeco/CNT Fiji) at a substrate temperature of 250 °C, using trimethylaluminium (TMA) and H₂O as precursors. The AlOₓ layer was then patterned via photolithography and reactive ion etching (CHF3: 15 sccm, Ar: 5 sccm, RF: 50 W, ICP: 300 W, Press: 7 mTorr). Next, the top electrodes, 500 nm wide and orthogonal to the bottom electrodes, were deposited using electron-beam lithography, followed by electron-beam evaporation of Ag (10 nm)/Au (50 nm) and lift off. The Ag (10 nm)/Au (50 nm) top electrode was created with a Lesker PVD-250 e-beam evaporator at a base pressure in the low 10⁻⁸ Torr range. The substrates were rotated at 20 rpm while kept at room temperature utilizing a chilled-water cooling stage. The system was equipped with a QCM feedback control to maintain the desired deposition rates within 3% tolerance. The device's active area (500 nm × 500 nm) corresponds to the area of cross-sectional overlap between the top and bottom electrodes. Finally, Ti (20 nm)/Au (200 nm) probe-contacts (100 μm × 100 μm) were deposited via photolithography and electron-beam evaporation.

The ARS was packaged in a commercial TO-5 can, and the electrodes were wire bonded to the package pins. Since the distance between package pins is several millimeters, the stray capacitance of the ARS package can be ignored.

The current–voltage characteristics of the ARS were measured by a Keysight B1500A semiconductor parameter analyzer.

**Avalanche pulse shape and quenching measurements**. As shown in Fig. 1d, the Si SPAD (Hamamatsu S14643-02) is connected in series with a quenching resistor and driven by a DC voltage source (Keithley 2400). A bias tee (ZFBT-4R2GW+) is used to extract the AC signal from the SPAD output. The bias tee has three ports, i.e., DC+AC input port (port ① in Fig. 3), AC output port (②), and DC output port (③). In the experiment, the SPAD, quenching resistor, 50 Ω readout resistor, and SMA type I/O port are soldered onto a PCB board. The 50 Ω readout resistor is used to match the PCB board impedance to the following circuits, and the bias tee was used to extract the avalanche pulse (port ②) from the DC background (port ③) and protect the amplifier and oscilloscope in case there is a constant and large current coming out from the PCB board (i.e., the SPAD is shorted). The avalanche pulse is then introduced into a low noise amplifier (ZFL-1000LN+) and read out using an oscilloscope (Rigol DS7024). A 520 nm laser (Thorlabs L520P120) driven by a Keysight 33600 A waveform generator delivers the light pulse to the SPAD. The Si SPAD's responsivity at 520 nm (0.2 A/W at a bias of 20 V with gain = 1) enables calibration of the input light intensity using the photo-current read by the Keithley 2400. The laser drive voltage is carefully set so that the corresponding photon number in each pulse averages to ~1000. The laser pulse is then attenuated to 1 photon/pulse by a ×1000 attenuator (Thorlabs NDUV530B). In this work, the current flowing through the SPAD is derived from the voltage readout (at the oscilloscope) divided by the voltage gain (10) of the low noise ZFL-1000LN+ amplifier times the AC port output impedance (50 Ω). The avalanche pulse shape studies were carried out with the laser repetition rate of 1 MHz and with the SPAD response recorded at a scanning step of 0.4 ns.

The over-voltages for ARS quenching and conventional quenching are carefully adjusted so that the single-photon detection rates for the SPAD quenched by the ARS and fixed quenching resistor (60 and 100 kΩ) are the same under a repetition rate of 1 MHz. The interval between two photons is long enough for the SPAD quenched by fixed resistance to have a full recovery. For the ARS-based dynamic quenching, since the ARS has a turn-on voltage of 8 V, the counting becomes significant only when the overvoltage is larger than 8 V. In this work, the overvoltage is taken to be 9 V. For conventional passive quenching, an overvoltage of 4 V is adequate to provide the same detection rate. The counting rate is estimated by reading the oscilloscope response curve and confirmed by connecting the AC output signal into a pulse counter (PicoHarp 300).

**Quenching measurement—SPAD response to high repetition rate photons**. Counting rates for the 20 MHz single-photon pulse repetition rate were measured over 8000 single-photon pulses. The data accumulated over 0.4 ms length, in time, with a scanning time step of 0.4 ns. Counting is achieved by setting a trigger threshold for the SPAD response trace[51]. The counting principle is shown in Fig. s2, which is similar to that used as a commercial counter (Picoharp 300). The thresholds for counting are chosen to be above the noise floor (see literature for instance[51]). For the 100 kΩ quenched SPAD, the threshold is chosen to be 40 mV, and the light counting rate is 1.8 MHz. For ARS quenching, the threshold is chosen to be 5 mV, and the light counting rate is 9.3 MHz.

**Counting histogram measurement and single-photon detection performance calculation**. A counter (HydarHarp 400) is used to replace the oscilloscope in Fig. 1d. The signal threshold and acquisition resolution are set to be 10 mV and 32 ps, respectively. The photon number in each pulse is attenuated to 0.1.

The photon detection efficiency is calculated from the total count probability ($P_t$) and dark count probability ($P_d$). $P_t$ and $P_d$ are defined as the avalanche pulse numbers per second divided by repetition rate, with and without light, respectively. The number of photo-generated e–h pairs ($n$) during each pulse obeys Poisson distribution ($f(n)$) and can be represented by[52]

$$f(n) = \frac{\lambda^n e^{-\lambda}}{n!} \tag{5}$$

Where $\lambda$ is the average number of photon-generated e–h pairs per pulse and is equal to $\eta \cdot \bar{n}$. $\eta$ is the quantum efficiency of the SPAD, and $\bar{n}$ is the average number of photons per pulse (0.1). Assuming the avalanche probability ($P_a$) is the same between the avalanche events triggered by each laser pulse. And[52]

$$P_a = 1 - (1 - P_d)(1 - P_b)^n \tag{6}$$

Where $P_b$ is the breakdown probability. Then the average avalanche probability per pulse, $P_t$, can be written as[52]

$$P_t = \sum_{n=0}^{\infty} P_a f(n) = 1 - (1 - P_d)^{-\bar{n} p_b} \tag{7}$$

Therefore, the SPDE of the SPAD, which is equal with $\eta \cdot p_b$, can be expressed as[52]

$$SPDE = \frac{1}{n} ln\left(\frac{1 - P_d}{1 - P_t}\right) \tag{8}$$

**PSPICE simulation**. The software OrCAD Pspice Designer was used to simulate the quenching process. The circuit schematic is shown in Fig. s3. The photon signal port, resistances $R1$, $R2$, and the switches $S_{Trig}$, $S_{Self}$ represent the switch in Fig. 1a–c. V1 and R3 represent the equivalent internal voltage source (breakdown voltage) and the SPAD internal resistance, respectively. $C1$ represents the SPAD junction capacitance. The optical switch sub-circuit, $V1$, $R3$, and $C1$ form the equivalent circuit of the SPAD. The quenching resistance is represented by $R4$ (ARS with PSPICE model embedded in). $V2$ is the external voltage source. $R5$ is the 50 Ω matching resistor. Components $C4$, $R6$, and $L1$ form a bias tee, which separates the AC and the DC signal. The values of $R6$ are given by the datasheet of the bias tee ZFBT-4R2GW+, where the values of capacitance and inductance are missing. Thus, $C6$ and $L1$ are selected from the datasheet of another bias tee product BT1-0026 from Marki Microwave, which has a similar transmission band to what we used in the experiment. The AC signal is introduced from $C4$ into an oscilloscope, whose input impedance is 50 Ω ($R7$). In the simulation, we track the current flow through $R7$, the voltage across SPAD, and the voltage and current on the ARS during quenching. The photon signal port generates a voltage pulse[53] with a pulse width of 1 ps whose rising edge triggers the switching (closure) of a voltage-controlled switch $S_{Trig}$. When $S_{Trig}$ switches on (i.e., closes), $C1$ discharges through an internal loop (labeled blue in the figure: $C1 \rightarrow R3 \rightarrow V1 \rightarrow S_{Trig} \rightarrow S_{Self} \rightarrow C1$). The discharge current exceeds the threshold of the current-controlled switch $S_{Self}$, leading to its closure when discharging begins. The falling edge of the electric pulse leads to the reopening of the voltage-controlled switch $S_{Trig}$. The current-controlled switch threshold is set to be 100 μA (latching current of self-sustainable avalanche[1]) in this work. When discharging ends, the current flow through $S_{Self}$ equals the excess bias (the difference between external voltage and breakdown voltage) divided by the total resistance (the sum of the quenching resistance and diode resistance). If the current is below 100 μA, $S_{Self}$ opens, and the avalanche continues unquenched. Else, the avalanche is quenched.

The SPAD breakdown voltage $V1$ is 100 V, and junction capacitance $C1$ is 0.7 pF (Hamamatsu S14643 datasheet). A typical value of 1 kΩ[54] is taken for the internal resistance, and with $V2=109$ V, the excess bias is 9 V. The recharging path is labeled in Fig. s3 as a red loop through $V1 \rightarrow R5 \rightarrow C1 \rightarrow ARS \rightarrow V1$.

A Pspice model of the ARS was built using an approach based on Biolek's work (Model R.2: Bipolar memristive system with threshold)[55], using the following equations to describe the behavior of the ARS:

$$I = x^{-1} V_M \tag{9}$$

$$\frac{dx}{dt} = f(V_M) W(x, V_M) \tag{10}$$

$$f(V_M) = \beta \times \left[ V_M - \frac{1}{2}(V_{on} + V_{off}) - \frac{1}{2}(|V_M - V_{off}| - |V_M - V_{on}|) \right] \tag{11}$$

$$W(x, V_M) = \theta(V_{on} - V_M)\theta\left(1 - \frac{x}{R_{off}}\right) + \theta(V_M - V_{off})\theta\left(\frac{x}{R_{on}} - 1\right) \tag{12}$$

Here $I$ and $V_M$ are the current and voltage on the ARS, $x$ is the resistance of the ARS, $\beta$ denotes a resistance transition speed (the unit is Ω/(s·V)). $V_{on}$, $R_{on}$, $V_{off}$, and $R_{off}$ are switch on voltage, on-state resistance, switch off voltage, and off-state resistance. Extracting the key parameters of the ARS from Fig. 4b, the $V_{on} = 8$ V, $V_{off} = 5$ V, $R_{on} = 40$ kΩ, and $R_{off} = 400$ kΩ. The response varies as a function of $\beta$, and it is found that the switching speed is greatly influenced by the factor $\beta$. $\beta$ looks to be in the $1 \times 10^{14}$ Ω/(sV) ballpark to be able to show a similar response to our experimental results. In the paper, $\beta$ is assumed to be $1 \times 10^{14}$ Ω/(sV) to accommodate the ~ns level rising and falling speed of the response curve. As described in Biolek's work[55], $\theta$ is the smoothed step function as shown in Eq. (13), which is set to avoid convergence problems[55]

$$\theta(x) = \frac{1}{1 + e^{-x/b}} \tag{13}$$

$$|x| = x[\theta(x) - \theta(-x)]. \tag{14}$$

Here, $b$ is a smoothing parameter ($b = 1 \times 10^{-5}$ according to Biolek's work[55]). Equation (14) defines the absolute value function by using the step function[55].

## Data availability

The data generated in this study have been deposited in the following Figshare database without accession code [https://doi.org/10.6084/m9.figshare.19092749.v1][https://doi.org/10.6084/m9.figshare.19092761.v1][https://doi.org/10.6084/m9.figshare.19092764.v1][https://doi.org/10.6084/m9.figshare.19092767.v1][https://doi.org/10.6084/m9.figshare.19092770.v1][https://doi.org/10.6084/m9.figshare.19092785.v1][https://doi.org/10.6084/m9.figshare.19092788.v1][https://doi.org/10.6084/m9.figshare.19092791.v1][https://doi.org/10.6084/m9.figshare.19092794.v1][https://doi.org/10.6084/m9.figshare.19092797.v1].

## Code availability

The codes used in this study have been deposited in the following Figshare database without accession code [https://doi.org/10.6084/m9.figshare.19114688.v1].

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

## Acknowledgements

J.Z., X.X., Y.Y., K.S., and L.W. would like to thank Junyi Gao and Xiayang Hua for a detailed discussion of experimental measurement. J.Z. and S.G. would like to thank Dr. Yeghishe Tsaturyan and Professor David D. Awschalom for a discussion on single-photon detection performance. S.G. acknowledges support from the Vannevar Bush Fellowship under the program sponsored by the Office of the Undersecretary of Defense for Research and Engineering and in part by the Office of Naval Research as the Executive Manager for the grant. The use of the Center for Nanoscale Materials, an

Office of Science User Facility, was supported by the U.S. Department of Energy, Office of Science, Office of Basic Energy Sciences, under Contract No. DE-AC02-06CH11357. The degradation experiment was supported by the National Natural Science Foundation of China (62175126) and the Ministry of Science and Technology of China under contract Nos. 2021ZD0109900 and 2021ZD0109903.

## Author contributions

J.Z. led the work. J.Z. was responsible for the thin film growths and ARS device fabrication (with D. Rosemann). J.Z. and K.S. designed the low noise single-photon detection system. J.Z., X.X., Y.Y., and K.S. were responsible for setting up the experiments and performing the measurements. J.Z. carried out the data analysis and circuit simulations, with C.J. assisting. J.Z., L.W., and J.W. were responsible for the SPAD degradation measurements. J.Z., S.G., and J.C.C. contributed to the writing of the paper, with S.G. and J.C.C. providing input on the materials and device aspects of the project, respectively.

## Competing interests

The authors declare no competing interests.
