## [Peer Review File · Nature Communications]

REVIEWER COMMENTS

Reviewer #1 (Remarks to the Author):

The authors have demonstrated a novel way for passive quenching in SPAD. The results look promising, however the reviewer still has some concerns as follows:

1. The authors should clearly address the difference between this submitted work and their previous work " Jiyuan Zheng, Xingjun Xue, Yuan Yuan, Keye Sun, Cheng Ji, Daniel Rosenmann, Joe Campbell, Supratik Guha, "Quenching of single photon avalanche photodiodes with dynamic resistive switches," Proc. SPIE 11721, Advanced Photon Counting Techniques XV, 117210E (12 April 2021) "in the introduction of revised manuscript.
2. There is obvious drift in the transfer curves of adaptive resistor. What's this influence on the afterpulsing, SPDE, DCR, jitter of SPAD after long-term operation (several hours maybe) as compared to the fixed resistor reference. These results are very important for practical applications of demonstrated technology. The authors must address this issue carefully in the revised manuscript. In fact, the ReRAM has been demonstrated over 20 years, however, it is still in the Lab. till now.
3. In order to further enhance the performances of SPAD, to operate it under cooling temperature is usually preferred. What's the performance of demonstrated adaptive resistor at different temperatures ? The authors had better to address such issue in the revised manuscript.
4. In Figure 4 (c), why the reference are not 400 kOhm for fair comparison ? The authors had better to provide this data in the revised manuscript.
5. In Figure 2 (a), can the authors provide measured jitter instead of a temporary waveform ?
6. As the authors claimed in the "discussion" section, the major contribution of such device is its low probability of afterpulsing. The authors should provide such data in the revised manuscript.

Reviewer #2 (Remarks to the Author):

In the present manuscript, the authors discuss the opportunity to improve the dynamic quenching of a single-photon avalanche photodetector using a volatile memristive switch. The work is motivated by the trade-off between the charging and discharging process requiring large and small R values. Here a volatile memristive switching devices is used, which is supposed to switch to the ON state and then facilitate a fast recharging process. If this process is mostly completed the volatile memristor will switch back to a high resistive state. The authors show some switching properties of the volatile devices and demonstrate fast quenching using the memristor in series with the SPAD device. By comparing the experimental data with simple circuit simulation including a behavioral memristor model the authors try

to proof that the proposed circuit outperforms standard quenching techniques. Using a volatile memristor to improve the dynamic quenching of a SPAD device seems to be a new idea. There are, however, some issues with the presented manuscript which need to be resolved. The following comments need to be addressed.

- 1) The authors state that memristive devices can switch in the sub-ns regime. This is, however only accomplished in specialized setups using coplanar wave guide structures. Here, the authors seem to connect the devices on a PCB, which should limit the speed significantly. The authors need to proof the switching speed of the fabricated devices. The results in Fig. 2c and 2d are obtained on a completely different time scale. Please note that the switching kinetics of these devices are highly nonlinear. Thus, a higher voltage needs to be applied to switch faster.
- 2) In the quenching circuit, the authors use very high voltages. The switching voltages of the volatile memristor are a lot lower as can be seen in Fig. 2. At these voltages, the memristive devices is likely to be destroyed. Indeed the authors show in 2d a degraded device characteristic.
- 3) The authors mention that the device switched up to 10^{10} times (estimated). Thus, there should be a large statistic for the recovery time. 2b, however, shows a lot less number of recovery times.
- 4) Fig. 2c shows also some switching variability, which is typical for these type of devices. Why is there only one cycle shown in 2d? The authors should provide more cycles after the quenching experiments to prove that the device is still working reliably.
- 5) The authors compare the recovery time of the ARS quenched SPAD and a fixed resistor quenched SPAD. The latter shows a longer recovery time, but it also has a larger resistance than the ARS ON state. The authors should use the same resistance value. For me it is not really clear that the device is really switching, but maybe just broke down more or less permanently.
- 6) To prove the concept the authors employ SPICE simulations using a very simple memristor model. This model does not capture the high nonlinearity of the switching dynamics, which is typically observed in this type of device. The authors should try to fit the experiment in 2c on the correct time scale (probably seconds) and then try to fit with the same parameters the switching in the nanosecond regime. Based on the equations, this will certainly fail completely.

Reviewer #3 (Remarks to the Author):

The article is well written and explains in sufficient detail a novel approach to SPAD quenching, with the potential of combining the simplicity of a passive quenching with a faster recovery time, conventionally only possible in circuits with active components.

For the sake of transparency, I have only commented on the parts of this paper that fall within my field of expertise, i.e. the SPAD operation, quenching and characterization, because the details regarding the ARS fabrication and operation mode fall outside of the topics that I can honestly review.

The paper is missing some measurements typically associated with SPAD devices and quenching circuits, such as a measurement of afterpulsing probability, but the authors are reasonably clear about the limitations of the ARS they were able to manufacture. This is also acceptable given that the authors claim that this work is a demonstration, ad not a final product.

I would like to suggest the authors to possibly add a small section describing whether they believe that this approach could be effectively scaled to SPAD arrays, for example Silicon PhotoMultipliers, which are

currently the largest application of passive quenched SPADs and would benefit significantly of a faster SPAD recharge.

Below I provide some comments about possible minor improvements to be made to this paper.

Page 4:

The discussion regarding the SPAD recharge current waveform, although supported by simulation results, lacks some experimental validation. For example, it would be useful to compare the SPAD recharge current waveform with your ARS to the same SPAD being recharged by a low-value resistor (comparable to the on-resistance of your ARS) and with a high-value resistor (comparable to the off-resistance of the ARS). This could help validate the explanation provided regarding the current waveform shape.

However, this might prove to be challenging from an experimental point of view, due to the low-value resistance possibly not enough to quench the SPAD, and the high-value resistance providing a current too close to the noise floor of the measurement. If this is indeed the case, one simpler figure of merit that could prove the effectiveness of the quenching behaviour of the ARS might be to report the cumulative avalanche charge flowing through the device, both with your ARS and with the conventional passive quenching resistor of the value used in this paper.

I believe that the faster quenching offered by your ARS should reduce the total avalanche charge and thus reduce the likelihood of afterpulsing in a SPAD.

Page 7, line 174:

The point that you mention here and explain further in “Methods - Avalanche pulse shape and quenching measurements” should probably be stated more explicitly, possibly at the beginning of section “Comparison with conventional passive quenching”. Indeed, it should be made clearer to the reader that the comparison between the ARS and the PQC has been made at different excess bias voltage, that were chosen such that they offered similar detection rates.

** See Nature Research’s author and referees’ website at www.nature.com/authors for information about policies, services and author benefits.

REVIEWER COMMENTS AND RESPONSES

Reviewer #1 (Remarks to the Author):

The authors have demonstrated a novel way for passive quenching in SPAD. The results look promising, however the reviewer still has some concerns as follows:

1. The authors should clearly address the difference between this submitted work and their previous work " Jiyuan Zheng, Xingjun Xue, Yuan Yuan, Keye Sun, Cheng Ji, Daniel Rosenmann, Joe Campbell, Supratik Guha, "Quenching of single photon avalanche photodiodes with dynamic resistive switches," Proc. SPIE 11721, Advanced Photon Counting Techniques XV, 117210E (12 April 2021) "in the introduction of revised manuscript.

Response: The SPIE submission noted by the referee was only an abstract, which has now been cited in the introduction section of the revised paper, along with a description of the difference from the present work (Page 3, lines 61-63).

2. There is obvious drift in the transfer curves of adaptive resistor. What's this influence on the afterpulsing, SPDE, DCR, jitter of SPAD after long-term operation (several hours maybe) as compared to the fixed resistor reference. These results are very important for practical applications of demonstrated technology. The authors must address this issue carefully in the revised manuscript. In fact, the ReRAM has been demonstrated over 20 years, however, it is still in the Lab. till now.

Response: Detailed degradation measurements have now been conducted on the SPAD quenched by the ARS. A new batch of ARS devices was fabricated and tested at Tsinghua University for this purpose. These new results are shown in Fig. 5 in the revised manuscript. We note progressive degradation: these additional results and discussion have been added in the revised manuscript (Page 10, lines 253-265, and Page 11, lines 266-267 (Main text); Page 17, lines 435-447 (Methods)). A discussion about how to mitigate such degradation is provided in Page 11, lines 268-273. The original Fig. 5 has been replaced to include the new degradation performance results and inserted in the supplementary materials section.

3. In order to further enhance the performances of SPAD, to operate it under cooling temperature is usually preferred. What's the performance of demonstrated adaptive resistor at different temperatures ? The authors had better to address such issue in the revised manuscript.

Response: A discussion about temperature-dependent performance has been provided in the discussion section (Page 12, lines 293-303).

4. In Figure 4 (c), why the reference are not 400 kOhm for fair comparison ? The authors had better to provide this data in the revised manuscript.

Response: We have now added these measurements. Experiments under the same condition have been conducted on a SPAD quenched by 40 kohm and 400 kohm passive quenching resistors for a fair comparison. Fig. 4c has been changed from the original figure to include these new results. Page 8, lines 192-197 have been added to the revised manuscript:

5. In Figure 2 (a), can the authors provide measured jitter instead of a temporary waveform ?

Response: Jitter performance has been provided in the revised manuscript, and the results are now shown in shown in Fig. 2c in the revised manuscript; Additional discussions are added on Page 5, lines 116-121:

6. As the authors claimed in the "discussion" section, the major contribution of such device is its low probability of afterpulsing. The authors should provide such data in the revised manuscript.

Response: A discussion (Page 12, lines 304-313) regarding afterpulsing has now been provided in the revised manuscript.

Reviewer #2 (Remarks to the Author):

In the present manuscript, the authors discuss the opportunity to improve the dynamic quenching of a single-photon avalanche photodetector using a volatile memristive switch. The work is motivated by the trade-off between the charging and discharging process requiring large and small R values. Here a volatile memristive switching devices is used, which is supposed to switch to the ON state and then facilitate a fast recharging process. If this process is mostly completed the volatile memristor will switch back to a high resistive state. The authors show some switching properties of the volatile devices and demonstrate fast quenching using the memristor in series with the SPAD device. By comparing the experimental data with simple circuit simulation including a behavioral memristor model the authors try to proof that the proposed circuit outperforms standard quenching techniques. Using a volatile memristor to improve the dynamic quenching of a SPAD device seems to be a new idea.

There are, however, some issues with the presented manuscript which need to be resolved. The following comments need to be addressed.

1) The authors state that memristive devices can switch in the sub-ns regime. This is, however only accomplished in specialized setups using coplanar wave guide structures. Here, the authors seem to connect the devices on a PCB, which should limit the speed significantly. The authors need to proof the switching speed of the fabricated devices. The results in Fig. 2c and 2d are obtained on a completely different time scale. Please note that the switching kinetics of these devices are highly nonlinear. Thus, a higher voltage needs to be applied to switch faster.

Response: The reviewer is correct that PCB connections limit our single-photon response timing performance. However, we do not claim to obtain sub-ns switching in our measurements, nor do we need a sub-ns response time. We are simply noting the literature data [Manzel et al. ²⁵ and references [5-19] therein, for instance] from prior work that described sub-ns switching speeds in similar filamentary devices. Our response time, as the reviewer has noted, is limited by the PCB connections and is of the order of a few nanoseconds. This is adequate for establishing the results demonstrating our idea, i.e., the significantly faster performance of the ARS compared with the fixed resistor. We have clarified this point in the revised manuscript (Page 10, lines 247-252.).

2) In the quenching circuit, the authors use very high voltages. The switching voltages of the volatile memristor are a lot lower as can be seen in Fig. 2. At these voltages, the memristive devices is likely to be destroyed. Indeed the authors show in 2d a degraded device characteristic.

Response: We believe there is a misunderstanding here that requires clarification. As we have described when introducing the working principle of the SPAD (Fig. 1b&c), the external voltage mostly drops on the SPAD. The static state of the quenching system is that all the voltage is shared by the SPAD (optical switch is open). While, dynamically, the voltage from the SPAD can be shared partly by ARS when there are avalanche pulse responses. But the highest voltage drop on the ARS is over-voltage, which is generally only several volts in magnitude and cannot burn the device. What's more, as can be seen from our supplemental experiment (Fig.5), the ARS has not died even after the quenching system has been continuously operated for 1h 30min. Statements have been added in the discussion section (Page 12, lines 314-319) for clarifying this point:

3) The authors mention that the device switched up to 10^{10} times (estimated). Thus, there should be a large statistic for the recovery time. 2b, however, shows a lot less number of recovery times.

Response: A statistical analysis of all the 10^{10} pulses is impossible given the data saving speed of the oscilloscope. However we have now increased the avalanche pulse number from 200 to 1000 and redone the counting statistics with this expanded number of data points in the analysis. The results are not significantly different from the earlier, smaller (200 points) dataset. The revised manuscript has been modified accordingly (Page 5, lines 107&109) and the histogram of Figure 2b is modified to include the increased statistics.

4) Fig. 2c shows also some switching variability, which is typical for these type of devices. Why is there only one cycle shown in 2d? The authors should provide more cycles after the quenching experiments to prove that the device is still working reliably.

Response: The multiple cycle measurement data is now provided. A new batch of ARS was fabricated and tested at Tsinghua University. As is shown in the inner plots of Fig. 5 a-c, reliable hysteresis behaviors (Multi-loops) are obtained for the ARS before and after quenching the SPAD. Detailed discussions about the experiments and degradation phenomenon are provided on Page 10, Lines 253-265, and Page 11, Lines 266-273.

5) The authors compare the recovery time of the ARS quenched SPAD and a fixed resistor quenched SPAD. The latter shows a longer recovery time, but it also has a larger resistance than the ARS ON state. The authors should use the same resistance value. For me it is not really clear that the device is really switching, but maybe just broke down more or less permanently.

Response: The revised figures 3(b), (c) and (d), which now include data from 40, 60, 100 and 400 k-ohm fixed resistances we believe explain our conclusion. When the fixed resistances are 40 k-ohm or higher, the response time is ~ 300 ns or higher. When the resistance is 30 k-ohm or less, we observe inadequate quenching of the avalanche because of the low resistance (enough current keeps flowing into the the SPAD and charging up the junction capacitance before it can get discharged), and the SPAD response becomes unreliable. Therefore, using a passive resistor we are unable to obtain response times less than a few hundred ns with **any** particular fixed resistance—either the resistance is too high for fast recovery, or the resistance is too low that the SPAD does not get quenched reliably. In the case of the ARS, on the other hand, we are able to observe a 10X reduction in response time to ~ 30 ns. Our passive resistance data clearly indicates that this is not possible if the ARS simply had a fixed resistance, since we have bracketed our fixed resistance data going down to values at which point the avalanche cannot be quenched. Therefore we conclude that the dynamic switching of the ARS is leading to this lowered quenching resistance. This point has now been made in the manuscript Page 7, Lines 166-174.

6) To prove the concept the authors employ SPICE simulations using a very simple memristor model. This model does not capture the high nonlinearity of the switching dynamics, which is

typically observed in this type of device. The authors should try to fit the experiment in 2c on the correct time scale (probably seconds) and then try to fit with the same parameters the switching in the nanosecond regime. Based on the equations, this will certainly fail completely.

Response: Our linear model can fit the avalanche pulse curve quite well. However, as the reviewer has pointed out, it cannot fit the I-V sweep curve. We now discuss this in the manuscript, as shown on Page 13, lines 320-325. In addition, the failed fitting results have been shown and discussed in supplementary materials (Fig. s4).

Reviewer #3 (Remarks to the Author):

The article is well written and explains in sufficient detail a novel approach to SPAD quenching, with the potential of combining the simplicity of a passive quenching with a faster recovery time, conventionally only possible in circuits with active components.

For the sake of transparency, I have only commented on the parts of this paper that fall within my field of expertise, i.e. the SPAD operation, quenching and characterization, because the details regarding the ARS fabrication and operation mode fall outside of the topics that I can honestly review.

1) The paper is missing some measurements typically associated with SPAD devices and quenching circuits, such as a measurement of afterpulsing probability, but the authors are reasonably clear about the limitations of the ARS they were able to manufacture. This is also acceptable given that the authors claim that this work is a demonstration, and not a final product.

Response: Thanks for the understanding. In the revised manuscript, we have carried out an estimate of the afterpulsing performance. This is discussed on Page 12, lines 304-313 of the revised manuscript.

2) I would like to suggest the authors to possibly add a small section describing whether they believe that this approach could be effectively scaled to SPAD arrays, for example Silicon PhotoMultipliers, which are currently the largest application of passive quenched SPADs and would benefit significantly of a faster SPAD recharge.

Response: We have added this description in the discussion section of the revised manuscript (Page 13, lines 326-329).

3) Below I provide some comments about possible minor improvements to be made to this paper.

Page 4:

The discussion regarding the SPAD recharge current waveform, although supported by simulation results, lacks some experimental validation. For example, it would be useful to compare the SPAD recharge current waveform with your ARS to the same SPAD being recharged by a low-value resistor (comparable to the on-resistance of your ARS) and with a high-value resistor (comparable to the off-resistance of the ARS). This could help validate the explanation provided regarding the current waveform shape.

However, this might prove to be challenging from an experimental point of view, due to the low-value resistance possibly not enough to quench the SPAD, and the high-value resistance providing a current too close to the noise floor of the measurement. If this is indeed the case, one simpler figure of merit that could prove the effectiveness of the quenching behaviour of the ARS might be to report the cumulative avalanche charge flowing through the device, both with your ARS and with the conventional passive quenching resistor of the value used in this paper.

Response: As discussed in the paper, we used a simplified model for fitting the avalanche pulse response. The on-state and off state resistances for the ARS are set as 40 k Ω and 400 k Ω . In this paper, a conventional passive quenching experiment is done to verify that the ARS doesn't stay at the fixed low value or high value resistance. For a fair comparison, 40 k Ω , 60 k Ω , 100 k Ω , and 400 k Ω quenched SPADs are tested, and the result has been shown in Fig. 3b-d & Fig. 4c. A detailed description of the recovery time comparison is provided on Page 7, lines 166-174, the recovery time of fixed resistor quenched SPAD is at least 10 times larger than that of the ARS quenched SPAD. A discussion about counting speed comparison is provided on Page 8, lines 192-197. The counting speed is significantly slower than ARS quenched SPAD, especially when the repetition rate is high. In conclusion, the critical switching of the ARS facilitates the fast quenching.

4) I believe that the faster quenching offered by your ARS should reduce the total avalanche charge and thus reduce the likelihood of afterpulsing in a SPAD.

Page 7, line 174:

The point that you mention here and explain further in "Methods - Avalanche pulse shape and quenching measurements" should probably be stated more explicitly, possibly at the beginning of section "Comparison with conventional passive quenching". Indeed, it should be made clearer to the reader that the comparison between the ARS and the PQC has been made at different excess bias voltage, that were chosen such that they offered similar detection rates.

Response: We have incorporated these points about afterpulsing on Page 12, lines 304-313.

A statement about excess bias voltage differences has been made in part "Comparison with conventional passive quenching" on Page 6, lines 144-145.

REVIEWER COMMENTS

Reviewer #1 (Remarks to the Author):

The authors have replied all my questions in very detail and now the paper can be accepted.

Reviewer #2 (Remarks to the Author):

All my questions were answered. I only have too small remarks. The authors need to correct two author names of the referenced papers:

- 1) It should be Menzel instead of Manzel.
- 2) The last name of the lead author of 49 is Russo and not Ugo.

Reviewer #3 (Remarks to the Author):

The additional measurements carried out by the authors helped to improve the quality of the paper, but I still have some concerns about parts of the revised manuscript.

1) Page 5, lines 116+ : it is my understanding that the authors have computed the quenched SPAD's jitter from the oscilloscope acquisitions from a ~ 15 ns laser pulse attenuated to a mean photon number of 1. This is sub-optimal, as the laser pulse is quite large and the mean photon number is such that significant pile-up distortion will be present, partly suppressing the second peak of the waveform. However, the authors have added the set of measurements reported at page 10, lines 253+, where the measurement conditions are much more suitable for device characterization: a mean photon number of 0.1 is good to avoid pile-up distortion and the obtained reconstructed waveform should be faithful to the device behaviour. The authors could refer to these measurements when talking about the device jitter, since they show much better dynamic range and are more useful overall. Jitter should also be quantified by reporting its rms or FWHM value.

2) It is clear to me that the jitter measurements reported both in figure 2c and in figure 5 were actually carried out with a laser whose optical pulse duration is much smaller than the 15 ns reported in page 5, line 109 (from the graphs I could make an educated guess that the FWHM of the reported graph is probably 5 ns or less) but no remark about this can be found in the revised document or in the additional documents. Please explain.

3) Furthermore, an additional incongruence can be noticed between the reported Dark Count Rates at page 8, lines 199-200 (330 kHz reported here) and the values in the new experiment at page 10, line 258 (8 kHz reported here). The authors should explain the different values: are they using a different SPAD? Is the bias voltage different? Are they cooling the device in any way?

** See Nature Research's author and referees' website at www.nature.com/authors for information about policies, services and author benefits.

REVIEWER COMMENTS AND RESPONSES

Reviewer #1 (Remarks to the Author):

The authors have replied all my questions in very detail and now the paper can be accepted.

Reviewer #2 (Remarks to the Author):

All my questions were answered. I only have too small remarks. The authors need to correct two author names of the referenced papers:

1) It should be Menzel instead of Manzel.

Response: Done. The name has been modified on Page 3 line 60, Page 4 line 92, and Page 10 line 240.

2) The last name of the lead author of 49 is Russo and not Ugo.

Response: Done. The name has been modified on Page 12 line 312.

Reviewer #3 (Remarks to the Author):

The additional measurements carried out by the authors helped to improve the quality of the paper, but I still have some concerns about parts of the revised manuscript.

1) Page 5, lines 116+ : it is my understanding that the authors have computed the quenched SPAD's jitter from the oscilloscope acquisitions from a ~ 15 ns laser pulse attenuated to a mean photon number of 1. This is sub-optimal, as the laser pulse is quite large and the mean photon number is such that significant pile-up distortion will be present, partly suppressing the second peak of the waveform. However, the authors have added the set of measurements reported at page 10, lines 253+, where the measurement conditions are much more suitable for device characterization: a mean photon number of 0.1 is good to avoid pile-up distortion and the obtained reconstructed waveform should be faithful to the device behaviour.

The authors could refer to these measurements when talking about the device jitter, since they show much better dynamic range and are more useful overall. Jitter should also be quantified by reporting its rms or FWHM value.

Response: Done. The 0.1 photon/pulse experiment has been referred to on Page 5, lines 120-121 when describing the jitter shape of SPAD. The FWHM values of jitter have been quantified and reported on Page 5, lines 118-119, and Page 10, line 262-263.

2) It is clear to me that the jitter measurements reported both in figure 2c and in figure 5 were actually carried out with a laser whose optical pulse duration is much smaller than the 15 ns

reported in page 5, line 109 (from the graphs I could make an educated guess that the FWHM of the reported graph is probably 5 ns or less) but no remark about this can be found in the revised document or in the additional documents. Please explain.

Response: This is now clarified and explained on Page 5, lines 121-124 in the revised manuscript. In our measurements, the modulation bandwidth of the TO-packaged laser (Thorlabs L520P120) compresses the 15 ns pulse width of the drive waveform (voltage monitored as shown in Fig. 2a red curve). The actual current waveform is narrower than the electric pulse, as a result of which the jitter time is much shorter than the electric input pulse width of the laser (15 ns) .

3) Furthermore, an additional incongruence can be noticed between the reported Dark Count Rates at page 8, lines 199-200 (330 kHz reported here) and the values in the new experiment at page 10, line 258 (8 kHz reported here). The authors should explain the different values: are they using a different SPAD? Is the bias voltage different? Are they cooling the device in any way?

Response: The reason for the different dark counts is because we used different overvoltages in the two measurements. All other conditions remained the same. This is now clarified on Page 10, lines 246-248, and Page 10, lines 251-252.

REVIEWERS' COMMENTS

Reviewer #3 (Remarks to the Author):

The authors have answered all of my previous questions with sufficient detail and the paper can now be accepted.

** See Nature Research's author and referees' website at www.nature.com/authors for information about policies, services and author benefits